# EP3 Is an Independent Prognostic Marker Only for Unifocal Breast Cancer Cases

**DOI:** 10.3390/ijms21124418

**Published:** 2020-06-22

**Authors:** Alaleh Zati Zehni, Udo Jeschke, Anna Hester, Thomas Kolben, Nina Ditsch, Sven-Niclas Jacob, Jan-Niclas Mumm, Helene Hildegard Heidegger, Sven Mahner, Theresa Vilsmaier

**Affiliations:** 1Department of Obstetrics and Gynecology & Breast Center, University Hospital, Ludwig Maximilian University of Munich, Marchioninistraße 15, 81377 Munich, Germany; Alaleh.Zati@med.uni-muenchen.de (A.Z.Z.); Anna.Hester@med.uni-muenchen.de (A.H.); Thomas.Kolben@med.uni-muenchen.de (T.K.); Nina.Ditsch@uk-augsburg.de (N.D.); Sven.Mahner@med.uni-muenchen.de (S.M.); 2Department of Obstetrics and Gynecology, University Hospital Augsburg, Universität Augsburg, Stenglinstr. 2, 86156 Augsburg, Germany; 3Department of General, Visceral, Transplant, Vascular and Thoracic Surgery, Ludwig Maximilian University of Munich, Marchioninistraße 15, 81377 Munich, Germany; Sven.Jacob@med.uni-muenchen.de; 4Department of Urology, Ludwig Maximilian University of Munich, Marchioninistraße 15, 81377 Munich, Germany; Janniclas.Mumm@med.uni-muenchen.de; 5Department of Obstetrics and Gynecology & Breast Center, University Hospital, Ludwig Maximilian University of Munich, Maistraße 11, 80337 Munich, Germany; Helene.Heidegger@med.uni-muenchen.de (H.H.H.); Theresa.Vilsmaier@med.uni-muenchen.de (T.V.)

**Keywords:** breast cancer, focality, prostaglandin E2 receptor 3 (EP3), prognosis, unifocal, multifocal

## Abstract

The aim of this study was to evaluate the prognostic impact of prostaglandin E2 receptor 3 (EP3) receptor expression might have on the two different breast cancer entities: multifocal/multicentric versus unifocal. As the prognosis determining aspects, we investigated the overall- and disease-free survival by uni-and multivariate analysis. To underline the study’s conclusion, we additionally considered the histopathological grading and the tumor node metastasis (TNM) staging system. A retrospective statistical analysis was performed on survival related events in a series of 289 sporadic breast cancer (BC) patients treated at the Department of Obstetrics and Gynecology at the Ludwig–Maximillian’s University in Munich between 2000 and 2002. The EP3 receptor expression was analyzed by immunohistochemistry and showed to have a significantly positive association with breast cancer prognosis for both entities, although with major differences. Patients with unifocal BC with EP3 receptor expression showed a significant improved overall survival, in contrast to the patient cohort with multifocal/multicentric BC. In this group, EP3 expression revealed its positive impact merely five years after initial diagnosis. Underlining the positive influence of EP3 as a positive prognosticator notably for unifocal breast cancer, only this patient cohort showed favorable outcomes in staging and grading. Especially EP3 expression in unifocal breast cancer was identified as an independent prognostic marker for the overall survival, when adjusted for age, grading, and staging. Altogether, our results strengthen the need to further investigate the behavior of EP3 in breast cancer and understand why markers linked to inflammation show different effects on prognosis and clinicopathological parameters on each focality type.

## 1. Introduction

Based on GLOBOCAN estimates, breast cancer is the most frequent cancer among woman worldwide with about 1.7 million cases in 2012 and an upward trend with 2.1 million cases in 2018, [1,2]. Even, if taking geographic and economic differences in consideration, incidence rates for BC still far exceed those for other cancers.

Therapy concepts either follow a curative intent or a prolongation of survival and maintaining quality of life in metastatic BC. Endless studies in the last decades lead to an immense progress of nowadays therapy regimes depending on clinical tumor subtype: from conventional e.g., surgery and chemotherapy to endocrine and targeted anti-tumor therapies e.g., tamoxifen and human epidermal growth factor receptor 2 (HER2)-blockade [3]. Since HER2-blockade is an established therapeutic target, it led to an immense improvement of the prognosis for patients with HER2 positive BC. However, to continue reducing BC recurrence and mortality, especially for patients with triple negative BC the development of new-targeted therapies is currently an active field of research.

Besides the well-known risk factors for BC, e.g., late menopause, nulliparity, obesity, the long-term use of hormone replacement therapy in postmenopausal woman—chronic inflammation has lately been linked to tumor progression. Therefore, the understanding of the signaling pathway of eicosanoids and their role in tumorigenesis have come into focus of recent research.

Primary enzymes in the synthesis of eicosanoids are the two isoforms of Cyclooxygenase (COX) enzymes: COX-1 and COX-2. While COX-1 is ubiquitously expressed in human tissue, COX-2 expression is regulated by cytokines and growth factors during inflammatory response. Both COX-enzymes catalyze the reaction from arachidonic acid to prostaglandin (PG) H_2,_ as a precursor of many biological significant molecules. The further conversion to specific PG`s, including PG E_2_, is metabolized by specific PG synthases. PG E_2_ exerts its biological effects via four G-protein coupled receptors: EP 1–4 (Prostaglandin E_2_ receptor 1–4) [4]. The prostanoid receptors EP 1–4 are divided into groups based on the type of G protein, through which they evoke cellular responses. EP1 unfolds its effect through Ca^2+^ dependent activation by Gq proteins [5]. EP2 and EP4 are coupled to Gs (G stimulatory) proteins, whose activation stimulates the production of cAMP. Prostaglandin receptor 3 (EP3), which exists in eight isoforms, mostly is coupled to a Gi (inhibitory) protein, whose activation reduces the cAMP production [5,6]. When existing in another isotype EP3 also binds to a Gs (stimulatory) protein, which results in increased cAMP levels [7]. So far, authors hypothesize that the mediation via different G-protein coupled pathways might be an explanation for the diverse effects of EP3 on tumorigenesis [8].

Exploiting the data, it appears that there is a consensus about EP2 and EP4 receptor upregulation being associated with a more aggressive course of disease [9,10,11,12,13,14]. Contrary to EP1 and EP3 expression in BC, where study outcomes are not unanimous. Most interestingly is the fact that when looking at the gynecological cancers, EP3 expression either proves a positive effect on survival rates e.g., on BC [15] or a negative one, e.g., on ovarian [16] and cervical cancer [17].

The importance of focality regarding the BC aggressiveness has been well reviewed in the last decade. Still, there is no clear standard international definition, but the most common understanding of multifocality is two or more separate tumor loci in the same quadrant, whereas multicentricity is defined as two or more separate invasive tumors in more than one quadrant of the same breast. Various study outcomes confirmed an association of multifocality and multicentricity with a more progressive course of disease: higher rate of distant metastasis, local relapse and shorter survival [18], lymph node metastases [19] and higher mortality rates [20]. According to this, authors have discussed the focality as an important prognosticator for BC [18,19,20].

To the best of our knowledge, no sufficient data to date exists concerning prognostic relevance of EP3 in breast cancer relating to the cancers focality. Therefore, the present study examined the expression of EP3 receptor in unifocal versus multifocal sporadic breast cancer and its impact on clinicopathological parameters, recurrence, and survival. Our research aimed to serve as a scientific base for future specific EP3 targeted BC therapy adopted to the focality.

## 2. Results

### 2.1. Unifocal BC

The expression of EP3 showed a statistically significant difference in the Overall Survival (OS) for the unifocal BC patients. The Kaplan–Meier Curve revealed that an EP3 expression is statistically correlated with a better OS, which was additionally supported by the Log-Rank test with *p* = 0.007 for the unifocal group (Figure 1a). Indicating the powerful role of EP3 as progosticator for the OS in unifocal BC, Kaplan–Meier-analysis (Figure 1a) revealed that EP3 positive patients have a better OS from the date of initial diagnosis. The disease-free survival of unifocal BC patients was not significantly affected by EP3 expression, however a trend could be observed (*p* = 0.074), also visualized by Kaplan–Meier analysis and calculated with Log-Rank test. Considering the histopathological tumor grading by WHO and the Tumor Node Metastasis (TNM) staging of the unifocal BC patients, statistical analysis revealed that unifocal BC patients expressing the EP3 receptor showed more favorable tumor characteristics (Table 1). Box-plots visualized (Figure 2a), and Kruskal–Wallis tests calculated a *p*-value of 0.007, that the EP3 positive unifocal group of patients were more often staged pT1 than pT2-4. Additionally, this patient group had a significant lower risk for the presence of metastasis, also shown by box-plots (Figure 2b) and Kurskal–Wallis with a *p*-value of 0.015. Furthermore, regarding the involvement of regional lymph nodes (pN *p* = 0.004) and the histopathological grading by WHO (*p* = 0.000) the unifocal patient group was influenced by the expression of EP3. Underling these results, Cox-regression revealed the EP3 expression to be an independent prognostic marker for OS (HR 0.246, 95% CI 0.100–0.603, *p* = 0.002) in this patient cohort (Table 2).

### 2.2. Multifocal and Multicentric BC

Again, same statistical devices were used to clarify and interpret the collected data. Also for the patient group, being diagnosed with multifocal BC an EP3 expression showed to have a positive impact on OS. The Kaplan–Meier curve showed that multifocal BC patients, have a better overall survival when being EP3 positive, which was confirmed by the Log-Rank test with a *p*-value of 0.048 (Figure 1b). Contrary to the unifocal cohort, EP3 in multifocal/multicentric BC was not a good prognosticator from the beginning on. Having a more detailed look at both Kaplan–Meier Curves (Figure 1a,b) it strikes out, that EP3 in multifocal/multicentric BC (Figure 1b) shows its impact on the OS merely 5 years after initial diagnosis. Just like the unifocal BC patients, an EP3 expression in the multifocal BC group revealed no significant influence on the disease-free survival (*p* = 0.822). Regarding the TNM staging (pT *p* = 0.562, pN *p* = 0.089, pM *p* = 0.208) and histopathological tumor grading by WHO (*p* = 0.453) the multifocal patient group was uninfluenced by the expression of EP3, calculated with Kruskal–Wallis tests. Contrary to the unifocal group, the EP3 could not be identified as an independent prognostic factor, when conducting Cox-regression (HR 0.927, 95% CI 0.498–1.724, *p* = 0.810) in the multifocal group (Table 3).

## 3. Discussion

The aim of this study was to evaluate the prognostic impact of EP3 receptor expression on the two different BC entities: unifocal vs. multifocal BC. EP3 proved to have a positive prognostic influence on both BC entities, but with major differences. In unifocal BC EP3 receptor expression was a positive prognosticator for the OS, and furthermore, an independent one. In addition, patients in this cohort showed to have more favorable characteristics in regard to TNM staging. Contrary to the multifocal/multicentric cohort; although patients with EP3 receptor expression had an improved OS, it was dependent of other clinicopathological parameters. Also, no correlation between EP3 receptor expression and TNM grading and staging was found.

Multiple studies have already identified COX-2 overexpression and elevated PGE_2_ levels in many malignant human cancers to be significantly associated with tumorigenesis and disease progression involving a poor prognosis and a metastatic phenotype [21,22,23,24]. With this knowledge, selective COX-2 inhibitors (COXib) like Celecoxib and Rofecoxib were successfully used in a chemopreventive manner [25]. Data from many controlled clinical trials and real-life experiences with Palbociclib—a highly selective cyclin-dependent kinase 4/6 inhibitor (CDK4/6), modulating the COX2-pathway—demonstrated a significant improvement in the prognosis of metastatic BC [26]. These findings imply a promising role of COX-2 antagonists as a therapeutic as well as preventive target in BC. Limiting factor for admission to standard therapy regimes is the toxicity of those drugs, like neutropenia, resistance, and severe side effects on the cardiovascular system [24,26].

Further research is ongoing, to intervene in another step of the COX2-PGE_2_-EP 1–4 pathway. In various cancer types, PGE_2_ is made responsible for stimulating angiogenesis, cancer cell growth, invasion, migration, suppression of immunity, and conferring resistance to apoptosis, when binding to its EP receptors [27,28]. Experimental evidence has indicated that each EP receptor may have a unique function in BC tumorigenesis. In murine COX2-induced mammary tumors, EP1, 2, and 4 receptors were strongly induced compared to normal mammary gland, whereas the EP3 receptor was downregulated [10]. Exploiting the data, it appears that there is a consensus about EP2 and EP4 receptor upregulation being associated with a more aggressive course of disease [9,10,11,12,13,14].

Contrary to EP1 and EP3 expression in BC, where study outcomes are not unanimous.

While some studies have pointed out the pro-tumorigenic role of EP1 in BC through vascular endothelial growth factor C (VEGF-C) production [29,30], others found EP1 to have an anti-metastatic function [31,32].

In a previous study, EP3 was introduced as significant prognosticator for improved progression-free and overall survival in sporadic BC [15]. This is consistent with Chang et al., whose findings in 2004 may be interpreted that EP3 overexpression in BC is a protective factor [10]. In addition, Hester et al. indicated that EP3 improved the BC prognosis independently from established clinicopathological parameters [15]. In a following study Hester et al. observed that even, if EP3 receptor expression is clinically a positive prognosticator in BC, it might rather be explained by aspects like immunological factors than tumor cell biology [8]. Comparing these results to our data, there is a consensus about EP3 being a positive marker for the prognosis and for some clinicopathological parameters, but with distinct exceptions when divided into two groups in regard to the focality. Former study outcomes suggested that the EP3 receptor plays a role in angiogenesis and EP3 agonists stimulate cell migration dose dependently in Chinese hamster ovary cells [33]; however which role EP3 specifically plays in breast cancer has yet to be determined.

It would also be of crucial relevance to understand, why receptors associated with inflammation response either lead to a weakened or more aggressive course of disease.

More recently, using the same patient cohort, we demonstrated that the prognostic value of hormone receptor expression e.g., estrogen-, progesterone-, and Vitamin-D receptor differs according to tumor focality [34]. That in fact opened a new perspective on the prognostic importance of receptor expression in BC, if valued without considering the cancers focality.

Therefore, we suggest enlarging the studies about Prostaglandin-mediated inflammatory reactions. By improving the understanding of the functional aspects of EP3 and its regulated factors, we aim to evaluate its eligibility as a possible future target not only in breast cancer prevention and treatment but also in all other gynecological cancer entities.

## 4. Materials and Methods

### 4.1. Patients

In this study, 289 breast cancer patients treated at the Department of Gynecology and Obstetrics at the Ludwig–Maximillian’s University from 2000 to 2002 were investigated. They all underwent BC surgery and were diagnosed with sporadic BC. Patients with a family history or distant metastasis were excluded from analysis. As the studies aim was to further research the prognostic differences between unifocal and multifocal and/or multicentric BC, the total collective (Table 4) was subdivided into two groups. Group 1 containing 173 patients being diagnosed with unifocal BC and group 2 made up of 147 patients diagnosed with multifocal as well as multicentric BC. The focality was diagnosed by clinical examinations, ultrasound and X-ray. In a few cases, in which additional information was necessary for a certain diagnosis of the focality, further techniques such as nuclear magnetic resonance imaging (NMRI), pneumocystography, or galactography were added.

The Institute of pathology of the Ludwig–Maximilian University of Munich assigned the histological type and the tumor grading by WHO (according to the Elston–Ellis system [35]); according to the union for international cancer control (UICC) TNM classification the tumor stage at primary diagnosis was classified [36]. From the Munich Cancer Registry, we retrieved the patient data such as hormone receptor status, HER2-amplification, patient age, metastasis, local recurrence, survival and progression. After an observation period of up to 10 years disease-free and overall survival was statistically analyzed.

### 4.2. Ethics Approval and Consent to Participate

The tissue samples used in this study were left over material after all diagnostics had been completed and were retrieved from the archive of Gynecology and Obstetrics, Ludwig–Maximilian University, Munich, Germany. All patients gave their consent to participate in the study. All patient data were fully anonymized, the study was performed according to the standards set in the declaration of Helsinki 1975. The current study was approved by the Ethics Committee of the Ludwig–Maximilian University Munich, Germany (approval number 048–08). Authors were blinded from the clinical information during experimental analysis.

### 4.3. Immunohistochemistry

For the identification of the EP3 status, BC tissue samples were fixated in formalin and afterwards embedded in paraffin after resection. Immunohistochemistry (Figure 3a–c) was performed according to the previously published methods [15,37] described in brief below. Primary antibodies used for the staining were Anti-EP3 (polyclonal rabbit IgG, Abcam, Cambridge, UK). Polymer-method (ZytoChem Plus HRP Polymer System mouse/rabbit, Zytomed Systems Berlin, Germany) and the chromogen diaminobenzidine (Dako, Hamburg, Germany) were used for detection. Positive controls were performed with placenta tissue, for negative control the primary antibodies were replaced with normal serum. The distribution and intensity patterns of specific immunohistochemical staining were evaluated using the well-established semi-quantitative immune-reactive score of Remmele and Stegner [38]. To calculate this score, staining intensity (graded as 0 = no, 1 = weak, 2 = moderate, and 3 = strong staining) was multiplied with the percentage of positively stained cells (0 = no staining, 1 ≤ 10% of cells, 2 = 11–50% of cells, 3 = 51–80% of cells, and 4 ≥ 81% of cells stained) and examined by using a Leitz microscope (Wetzlar, Germany). Samples with an immune-reactive score of Remmele and Stegner (IRS) of 0 or 1 were defined as EP3 negative and samples with an IRS of >3 were defined as EP3 positive. Both groups were then compared for clinicopathological parameters such as Grading and disease-free and overall survival.

### 4.4. Statistical Analysis

Statistical analysis was performed using the computer software “Statistical Package for the Social Sciences” (IBM SPSS Statistic 24.0 Inc., Chicago, IL, USA). In this study *p*-values of less than 0.05 were considered statistically significant.

The TC database was divided due to the focality into two groups. Group 1, which contained the unifocal BC patients and group 2, consisting of the multifocal and/or multicentric BC patients. Differences between EP3 receptor positive and negative patients—always relating to the focality—influencing the prognosis were tested for significance. Kaplan–Meier Curve analysis was performed for each group to estimate the disease-free and overall survival in EP3 positive and negative patients. By applying the chi-square of the log rank test the significance was determined. As the device for statistical analysis, boxplots and Kruskal–Wallis tests were used. In order to evaluate whether EP3 expression is an independent prognostic factor, multivariate analyses via COX-regression were conducted. For multivariate analysis we included the following factors: tumor size, lymph node status, metastasis, grading, patient age, EP3, VDR, ER, and PR expression.

## 5. Conclusions

In conclusion, the present study analyzed the role of EP3 as a prognosticator for the two BC entities: multifocal/multicentric vs. unifocal BC and found highly significant differences.

While EP3 expression in unifocal BC showed a significant positive impact on prognosis determining factors, e.g., TNM staging and grading, EP3 expression in multifocal/multicentric BC did not influence the four analyzed set points at all. These findings were underlined by the results of the OS analysis. In unifocal BC, EP3 positivity showed a better OS from the initial diagnosis on, whereas in multifocal BC the positive impact of EP3 expression on the OS was not given before 5 years after initial diagnosis. Additionally, EP3 in unifocal BC proved to be an independent prognosticator regardless of age, staging and grading of the investigated patient cohort. In contrast to multifocal BC, where EP3 expression revealed to be not an independent marker. Regarding the role of EP3 in the disease-free survival (DFS) of the patients, no correlations were proven, neither for the unifocal, nor for the multifocal patient cohort.

Our findings strengthen the need to further investigate the behavior of EP3 in BC and enlarge the studies for the cause of BC focality type. To the best of our knowledge, this study is the first to investigate the correlation between EP3 and BC prognosis dependent on the focality. The importance and representativeness of our research are underlined by the large patient collective of 289, supported by the highly significant results. These data may contribute to novel progress and provide a promising perspective for innovation in the systematic therapy regimes for women with sporadic BC.

## Figures and Tables

**Figure 1 ijms-21-04418-f001:**
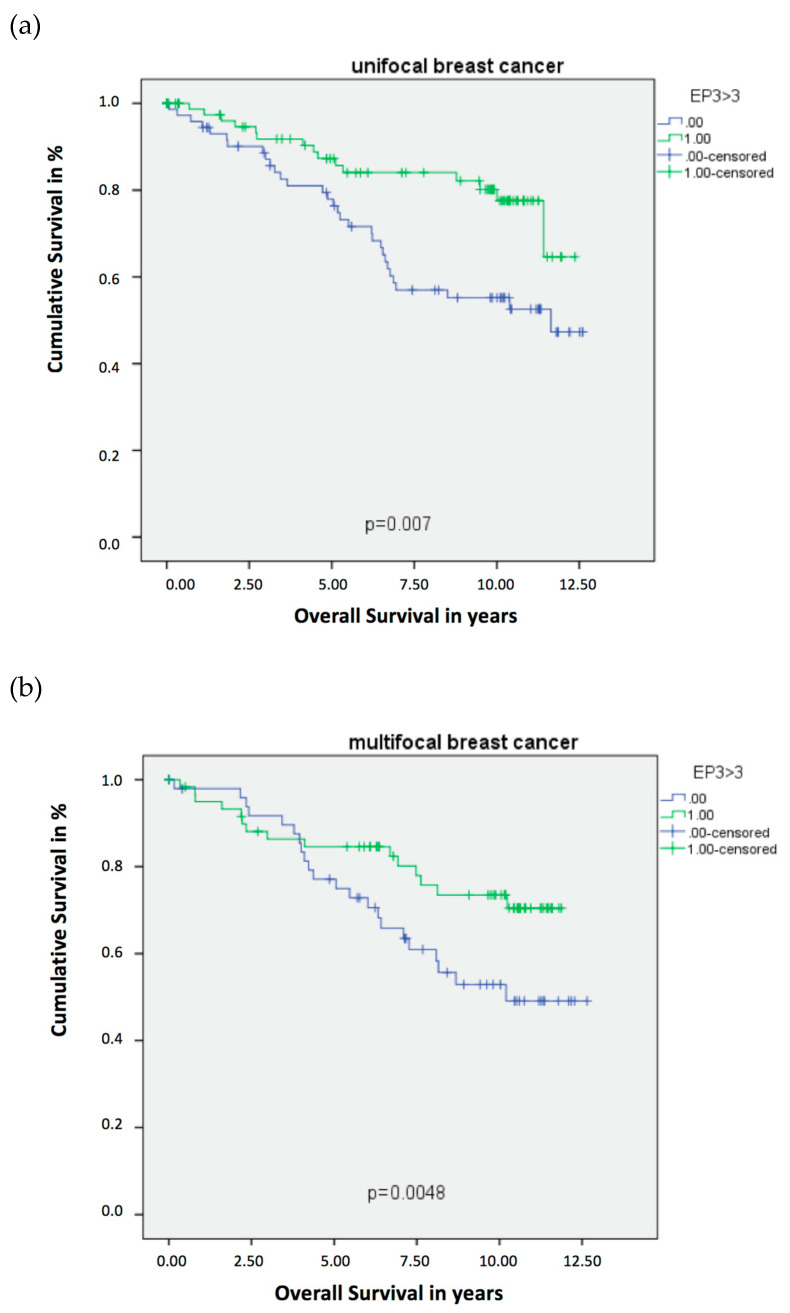
Kaplan–Meier survival analysis among EP3 positive and negative patients. Blue graph: EP3 receptor negative patient cohort (immune-reactive score of Remmele and Stegner (IRS) < 3). Green graph: EP3 receptor positive patient cohort (IRS > 3). (**a**) Overall Survival of patients with unifocal BC. (**b**) Overall Survival of patients with multifocal BC.

**Figure 2 ijms-21-04418-f002:**
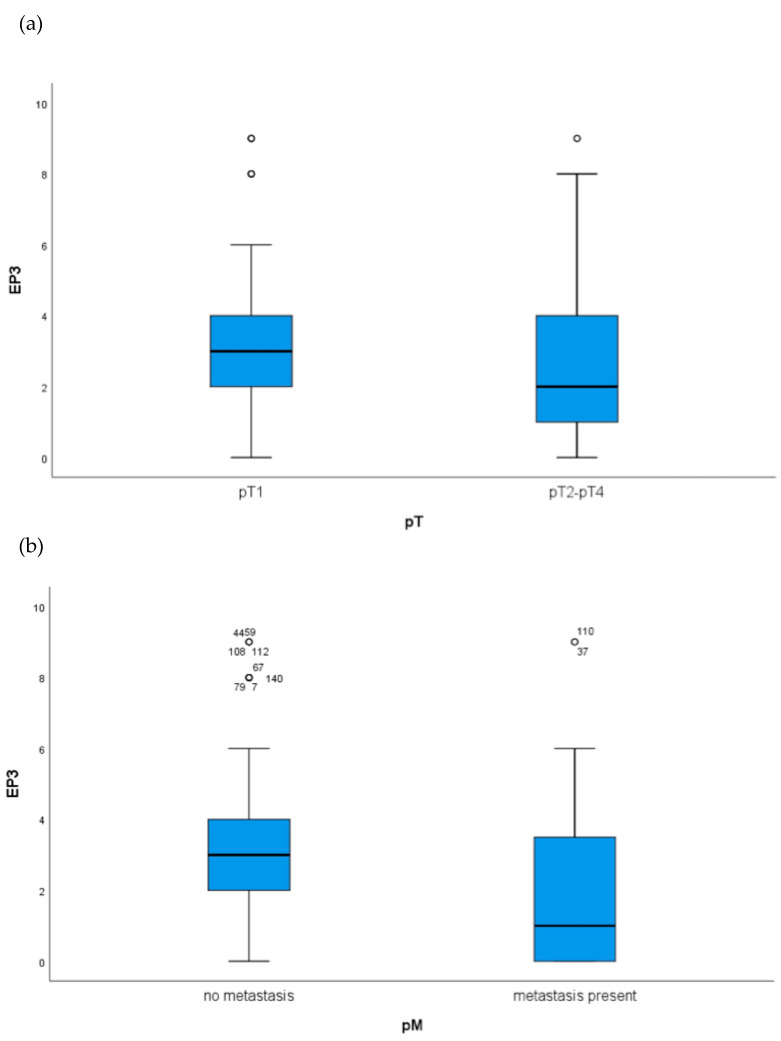
Boxplots analysis of the unifocal BC group in regard to tumor node metastasis (TNM)staging: (**a**) EP3 expression in unifocal BC patients in regard to T stage (EP3 positive unifocal group are more often staged pT1 than pT2-4 *p* = 0.007) and (**b**) M stage (EP3 positive unifocal group have a significant lower risk for metastasis *p* = 0.0015).

**Figure 3 ijms-21-04418-f003:**
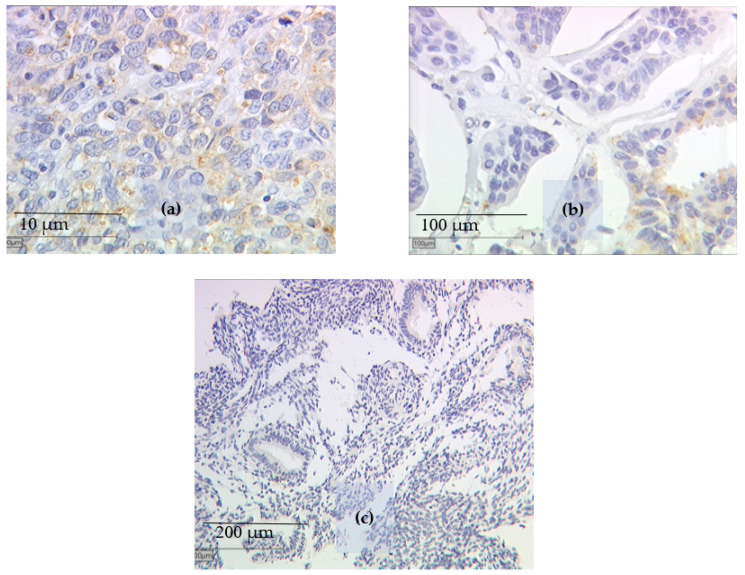
Immunohistochemical EP3 staining after incubation with the primary antibody of breast cancer tissue samples is displayed: (**a**) pT1 staged patient samples with an IRS > 3, (**b**) pT2-4 staged patient samples with an IRS < 2, and (**c**) EP3 negative sample in endometrial cancer.

**Table 1 ijms-21-04418-t001:** Significant results for the Prostaglandin E2 receptor 3 (EP3) receptor positive patients.

EP3	Unifocal	Multifocal
Overall survival	● +	● +
Disease-free survival		
Grading	● +	
pT	● +	
pN		
pM	● +	

● = Expression of the particular receptor has a significant influence of the marked characteristics. + = Receptor expression effects the marked characteristics significant positively.

**Table 2 ijms-21-04418-t002:** Multivariate Cyclooxygenase (COX) regression analysis of unifocal breast cancer (BC) patients.

Variable	Coefficient	HR (95%CI)	*p*-Value
Age	0.028	1.028 (0.997–1.061)	0.079
Grading	1.358	3.889 (1.899–7.964)	**0.000**
pT	0.297	1.346 (1.005–1.8804)	**0.046**
pN	0.644	1.904 (1.225–2.960)	**0.004**
pM	−1.190	0.304 (0.049–1.883)	0.201
EP3	−1.403	0.246 (0.100–0.603)	**0.002**
VDR	0.066	1.068 (0.912–1.251)	0.417
ER	−0.861	0.423 (0.151–1.184)	0.101
PR	−0.533	0.587 (0.231–1.489)	0.262

Significant results are shown in bold; HR: hazard ratio; CI: confidence interval.

**Table 3 ijms-21-04418-t003:** Multivariate Cox regression analysis of multifocal BC patients.

Variable	Coefficient	HR (95%CI)	*p*-Value
Age	0.031	1.032 (1.007–1.057)	**0.012**
Grading	0.002	1.002 (0.995–1.008)	0.591
pT	0.289	1.335 (1.030–1.730)	**0.029**
pN	0.320	1.377 (1.087–1.743)	**0.008**
pM	−0.306	0.737 (0.585–0.928)	**0.009**
EP3	−0.76	0.927 (0.498–1.724)	0.810
VDR	0.100	1.105 (0.940–1.299)	0.226
ER	−1.227	0.293 (0.98–0.874)	0.028
PR	−0.869	0.419(0.173–1.017)	0.055

Significant results are shown in bold; HR: hazard ratio; CI: confidence interval.

**Table 4 ijms-21-04418-t004:** Patient characteristics of the total collective.

Patient Characteristics	*n* (%)
Age (years)	Mean 59.9 Standard deviation 13.06
Tumor foci	Unifocal 151 (52.2) Multifocal 138 (47.8)
Histology	NST 144 (49.8) Non-NST 145 (50.2)
Tumor grade	G1 or G2 107 (70.9) G3 44 (29.1)
pT	pT1 193 (66.8) pT2-pT4 96 (33.2)
pN	pN0 165 (57.5) pN1-pN3 122 (42.5)
EP3 +	Unifocal 101 (51.8) Multifocal 94 (48.2)
EP3 −	Unifocal 50 (53.2) Multifocal 44 (64.8)

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
