# Peer review of "EP3 Is an Independent Prognostic Marker Only for Unifocal Breast Cancer Cases"

_ijms, 2020, doi:10.3390/ijms21124418_

Round 1

Reviewer 1 Report

The present study demonstrated that EP3 is an independent prognosticator in unifocal breast cancer. In my opinion, this manuscript is well written and the conclusion can be supported by the statistical result. However, the cohort have been analyzed in several published studies. To confirm whether EP3 is an independent prognosticator unifocal breast cancer, more evidences are required.

  1. The prognostic value of EP3 have been evaluated in the same cohort (reference 15 cited in this manuscript, published in BMC cancer, 2018, 18(1):431.). Therefore, evaluating EP3 expression in other cohort is more convincing.
  2. In addition, the authors demonstrated that prognostic value of hormone receptor expression and Vitamin-D receptor in unifocal breast cancer in the same cohort. The variables in the Table 2 and Table 3 include age, grading, TNM, and EP3. Did the authors try to perform multivariate Cox regression analyzes according to the expression of hormone receptors, Vitamin-D receptor, and EP3?
  3. Please enhance the resolution of figure 1 and figure 2, the labels are not clear enough.

Author Response

Comments and Suggestions for Authors

The present study demonstrated that EP3 is an independent prognosticator in unifocal breast cancer. In my opinion, this manuscript is well written and the conclusion can be supported by the statistical result. However, the cohort have been analyzed in several published studies. To confirm whether EP3 is an independent prognosticator unifocal breast cancer, more evidences are required.

1. The prognostic value of EP3 have been evaluated in the same cohort (reference 15 cited in this manuscript, published in BMC cancer, 2018, 18(1):431.). Therefore, evaluating EP3 expression in other cohort is more convincing.

1. Authors: Indeed the prognostic value of EP3 has been evaluated in the same patient cohort in sporadic breast cancer, showing that EP3 receptor expression is a significant prognostic factor for improved progression-free and overall survival. However, the functional aspects and prognostic impact of EP3 receptor in breast cancer has not been evaluated in relation to the two different breast cancer entities: multifocal/multicentric versus unifocal. In order to support out previous study, this paper aims at identifying the factors regulated by EP3, the same patient cohort was necessary to evaluate the possibility of targeting EP3 in future anti-tumor therapy in breast cancer. This was the reason for further immunohistochemical staining for specifially multifocal/multicentric versus unifocal BC in the same patient cohort. By using the same patient cohort we were able to underline the positive influence of EP3 as a positive prognosticator notably for unifocal breast cancer, only this patient cohort showed favorable outcomes in staging and grading. Especially EP3 expression in unifocal breast cancer was identified as an independent prognostic marker for the overall survival. Nevertheless, we do appreciate the advise for further evaluating EP3 expression in other patient cohort in the future for supporting our thesis.

2. In addition, the authors demonstrated that prognostic value of hormone receptor expression and Vitamin-D receptor in unifocal breast cancer in the same cohort. The variables in the Table 2 and Table 3 include age, grading, TNM, and EP3. Did the authors try to perform multivariate Cox regression analyzes according to the expression of hormone receptors, Vitamin-D receptor, and EP3?

2. Authors: Cox regression was furthermore included in the manuscript according to the expression of hormone receptors, Vit. D receptor and EP3 (see below).

Table 2. Multivariate Cox regression analysis of unifocal BC patients

Variable

Coefficient

HR (95%CI)

P Value

Age

0,028

1,028 (0,997-1,061)

0,079

Grading

1,358

3,889 (1,899-7,964)

0,000

pT

0,297

1,346 (1,005-1,8804)

0,046

pN

0,644

1,904 (1,225-2,960)

0,004

pM

-1,190

0,304 (0,049-1,883)

0,201

EP3

-1,403

0,246 (0,100-0,603)

0,002

VDR

0,066

1,068 (0,912-1,251)

0,417

ER

-0,861

0,423 (0,151-1,184)

0,101

PR

-0,533

0,587 (0,231-1,489)

0,262

Significant results are shown in bold; HR: hazard ratio; CI: confidence interval

Table 3. Multivariate Cox regression analysis of multifocal BC patients

Variable

Coefficient

HR (95%CI)

P Value

Age

0,031

1,032 (1,007-1,057)

0,012

Grading

0,002

1,002 (0,995-1,008)

0,591

pT

0,289

1,335 (1,030-1,730)

0,029

pN

0,320

1,377 (1,087-1,743)

0,008

pM

-0,306

0,737 (0,585-0,928)

0,009

EP3

-0,76

0,927 (0,498-1,724)

0,810

VDR

0,100

1,105 (0,940-1,299)

0,226

ER

-1,227

0,293 (0,98-0,874)

0,028

PR

-0,869

0,419(0,173-1,017)

0,055

Significant results are shown in bold; HR: hazard ratio; CI: confidence interval

3. Please enhance the resolution of figure 1 and figure 2, the labels are not clear enough.

3. Authors: Figure 1 a&b and 2 a&b have been fully revised in the manuscript (see below)

Figure 1. Kaplan-Meier survival analysis among EP3 positive and negative patients.                                       Blue graph: EP3 receptor negative patient cohort (IRS<3). Green graph: EP3 receptor positive patient cohort (IRS >3) (a) Overall Survival of patients with unifocal BC (b) Overall Survival of patients with multifocal BC

Figure 2. Boxplots analysis of the unifocal BC group in regard to TNM staging (a) EP3 expression in unifocal BC patients in regard to T stage (EP3 positive unifocal group are more often staged pT1 than pT2-4 p=0.007) and (b) M stage (EP3 positive unifocal group have a significant lower risk for metastasis p=0.0015)

Reviewer 2 Report

In this manuscript the Authors evaluated the role of the membrane-bound EP receptor specific for PGE2 subtype EP3 in breast cancer. They concluded that EP3 is a prognosticator for unifocal BC.

The study was well designed and the final conclusion is well supported by experimental data. The presentation of the results was clear in the most cases. I have no comments regarding immunohistochemical studies or statistical analysis, but I have few minor comments regarding other aspects:

  1. Table 2: Is the patient's age range really 69? The number 69 indicates the average age of the studied cohort.
  2. Please add the OS abbreviation extension.
  3. Figures 1 and 2 are of poor quality. Axes and units are barely visible. Very small font.
  4. Figure 3: bar scale is hardly visible. Only in photo B you can see at a high magnification that the scale corresponds to 100 μm. In the remaining photos the number 100 μm is out of the area.

Author Response

Comments and Suggestions for Authors

In this manuscript the Authors evaluated the role of the membrane-bound EP receptor specific for PGE2 subtype EP3 in breast cancer. They concluded that EP3 is a prognosticator for unifocal BC.

The study was well designed and the final conclusion is well supported by experimental data. The presentation of the results was clear in the most cases. I have no comments regarding immunohistochemical studies or statistical analysis, but I have few minor comments regarding other aspects:

Authors: Thank you for your helpful comments. We agree to the points raised. All Tables and Figures have been fully revised (see below and in Section 2.3 of the revised manuscript)

1. Table 2: Is the patient's age range really 69? The number 69 indicates the average age of the studied cohort.

1. Authors: We recalculated the age of our patient cohort. Mean 59.9 / standard deviation 13.06. The data was revised in the manuscript.

2. Please add the OS abbreviation extension.

2. Authors: We added the OS to the abbreviation extension.

3. Figures 1 and 2 are of poor quality. Axes and units are barely visible. Very small font.

3. Authors: Figure 1 a&b and 2 a&b have been fully revised in the manuscript (see below)

Figure 1. Kaplan-Meier survival analysis among EP3 positive and negative patients.                                       Blue graph: EP3 receptor negative patient cohort (IRS<3). Green graph: EP3 receptor positive patient cohort (IRS >3) (a) Overall Survival of patients with unifocal BC (b) Overall Survival of patients with multifocal BC

Figure 2. Boxplots analysis of the unifocal BC group in regard to TNM staging (a) EP3 expression in unifocal BC patients in regard to T stage (EP3 positive unifocal group are more often staged pT1 than pT2-4 p=0.007) and (b) M stage (EP3 positive unifocal group have a significant lower risk for metastasis p=0.0015)

4. Figure 3: bar scale is hardly visible. Only in photo B you can see at a high magnification that the scale corresponds to 100 μm. In the remaining photos the number 100 μm is out of the area.

4. Authors: Figure 3 (a-c) has been fully revised in the manuscript (see below).

Round 2

Reviewer 1 Report

Thanks for authors’ responses. I have no further comments.